# Parasitic Nematode and Protozoa Status of Working Sheepdogs on the North Island of New Zealand

**DOI:** 10.3390/ani9030094

**Published:** 2019-03-18

**Authors:** Adam O’Connell, Ian Scott, Naomi Cogger, Boyd R. Jones, Kate E. Hill

**Affiliations:** 1Working Dog Centre, School of Veterinary Science, Massey University, Private Bag 11 222, Palmerston North 4442, New Zealand; adam.oconnell@outlook.com (A.O.); b.jones@massey.ac.nz (B.R.J.); k.hill@massey.ac.nz (K.E.H.); 2Pre-Clinical, Imaging and Research Laboratories (PIRL), South Australia Health and Medical Research Institute (SAHMRI), Adelaide, SA 5000, Australia; 3School of Veterinary Science, Massey University, Private Bag 11 222, Palmerston North 4442, New Zealand; i.scott@massey.ac.nz

**Keywords:** working farm dogs, herding dogs, parasites, *Toxocara canis*, *Sarcocystis*, husbandry

## Abstract

**Simple Summary:**

Working farm dogs may be at more risk of infection with gastrointestinal parasites than pet dogs, as they are fed raw meat and are in close contact with other dogs. This study determined the percentage of working farm dogs in New Zealand shedding intestinal parasite stages in their feces and explored what factors might increase or decrease the chance of parasites being in a dog’s feces. One person collected information about the dogs and their management using a questionnaire, body condition scored each dog, and collected a fecal sample to test for parasites. The study found that four out of ten dogs had one or more types of gastrointestinal parasites present in their feces, and this was more common in younger dogs. There was no association between the presence of parasites in feces and frequency that owners reported giving dogs anthelmintic drugs. The high percentage of dogs with parasite lifecycle stages present in their feces is of concern for the health of the dogs and also the owners as some parasites are zoonotic.

**Abstract:**

Working farm dogs in New Zealand may have a high parasitic challenge because of access to raw meat and close contact with other dogs. This cross-sectional study aimed to estimate the percentage of dogs with gastrointestinal nematode and protozoan parasite lifecycle stages present in their feces and to identify factors associated with the presence of parasites. A single researcher collected information about the dogs and their management via a questionnaire, body condition scored (BCS) the dogs, and collected fecal samples to determine the parasite burden. Fecal samples were collected from 171 dogs and 40% (95% CI 33.0% to 47.7%) contained parasite ova or (oo)cysts. There was no association between BCS and the presence of nematodes and parasites (*p* = 0.74) in the feces. The percentage of dogs with parasites present in their feces was not associated with BCS or the frequency with which anthelmintic drugs were reportedly administered (*p* = 0.61). The high percentage of dogs with parasites are of concern for the health of the dogs and their owners, given the zoonotic potential of some parasites. Further, research should also focus on understanding why reporting giving anthelmintic drugs at least every three months did not eliminate the infection.

## 1. Introduction

The helminth and protozoan parasites of the canine gastrointestinal tract are a concern not only for the health of the host but also for their owners, given the zoonotic potential of some parasites [1]. Numerous surveys have examined the prevalence of gastrointestinal parasites in dogs across the world, (e.g., [1,2,3,4]). In New Zealand, dogs have been shown to be host to several helminth and protozoan parasites [5]. In the limited survey work done, *Toxocara canis*, *Trichuris vulpis*, and *Uncinaria stenocaphala* were the most common nematodes reported in dogs [6,7]. *Ancylostoma caninum* has been reported in New Zealand sporadically [8,9], but most reports of this parasite have been in racing greyhounds. In terms of cestodes, numerous *Taenia* spp. tapeworms have been described in New Zealand dogs as well as *Dipylidium caninum* [5,6,10]. New Zealand has been declared provisionally free of *Echinococcus granulosus* since 2002 [11], but other cestodes, especially *Taenia ovis*, are still common. Therefore, New Zealand sheep farmers are encouraged to treat their dogs with praziquantel monthly [12]. No trematodes have been described in dogs in New Zealand. In terms of protozoan species with cysts, oocysts or sporocysts shed in feces, *Cryptosporidium* spp., *Cystoisospora* spp., *Neospora caninum*, *Hammondia* spp., *Giardia* spp., and *Sarcocystis* spp. have all been reported [5].

In New Zealand, working farms dogs represent a subgroup of dogs that may encounter a high parasitic challenge due to their environment and husbandry. A survey of New Zealand Sheep Dog Trial members observed that 60% of working farms dogs are fed meat that had been killed and butchered on farm, commonly called home kill meat in New Zealand, typically in combination with dry food [13]. Farm dogs are usually run in teams that have an average of six dogs [13], and the dogs are often confined overnight in kennels with or near other dogs which may increase their exposure to parasites [14]. The risks posed by feeding home killed meat and kenneling can be mitigated somewhat by freezing or cooking the meat before feeding and a preventative health regime that includes appropriate use of anthelmintic drugs.

Unfortunately, none of the surveys involving New Zealand working farm dogs have captured detailed information of husbandry and management of working farm dogs [13,15]. The aims of this study were to (1) determine the prevalence of gastrointestinal nematode and protozoan parasites in working farm dogs based on the presence of parasite eggs or (oo)cysts in feces; (2) collect information as to the management and husbandry of the working dogs; and (3) determine if any dog factors (e.g., age, sex, and breed) or management factors (e.g., frequency kennels are cleaned) influenced the prevalence of parasite infection.

## 2. Materials and Methods

### 2.1. Overview

The study used a cross-sectional study design and data was collected by a single investigator (AO) between 11 April 2010 and 7 July 2010 that was approved by the Massey University Animal Ethics Committee, MUAEC 10/10. Data collection involved multiple components: (1) questionnaire completed by owners to obtain information about the dogs and current husbandry practices; (2) physical examination which included body condition scoring and a detailed eye examination; (3) collection of fecal sample to determine whether parasite lifecycle stages were present; and (4) collection of blood from a subset of dogs that appeared healthy on physical examination to establish reference intervals for hematology and biochemistry in working farm dogs. Not all owners and dogs contributed data to every section of the study (see Figure 1). This paper presents the results from the questionnaire, body condition scoring, and fecal analysis, and as such, the methods will not include details of the eye examination or collection and analysis of blood data. Follow-up papers will describe the results of the eye examination and explore reference ranges for hematology and biochemistry in working dogs.

### 2.2. Subject Selection

The criteria for inclusion into the study were that the dog owner and their dogs had to work on a sheep farm, beef farm or sheep and beef farm. Owners were recruited in two ways. Firstly, the lead author (AO) approached owners of working dogs who were clients of a single veterinary clinic in the Waikato region of the North Island of New Zealand and considered, based on the perceptions of the author and others in the veterinary practice, as likely to be willing to participate. If the owner agreed to join the study, then the lead author visited the farm, and all dogs on the farm were enrolled in the study if they were older than six months and still working with stock. Secondly, a group of owners were recruited at the Tux North Island Dog Trial Championship 2010, in Gisborne. Owners recruited at the dog trial only provided data for those dogs that were present at the trial.

### 2.3. Data Collection

The owner was asked to complete a two-part questionnaire (see Appendix A). The first section collected data on the breed, age, sex, and neuter status. The second section contained questions relating to husbandry and care of the dogs including feeding, kenneling, vaccination, and use of anthelmintic agents for intestinal parasites. The questions assumed owners managed all dogs in the same manner and owners were asked to leave the question unanswered if their practices varied between dogs. When the owner varied management practices between dogs, the lead author made relevant notes and recorded the differences when entering the data.

Each dog had a physical examination completed by the lead author, who was a registered veterinarian. The results of the physical examination were recorded in a standardized proforma (Appendix A). The examination included an assessment of the major body systems for presence or absence of abnormalities, detailed eye examination, and body condition score (BCS). Each dog’s body condition was assessed visually and after palpation assigned a rating using a validated nine-point scale with 1 being an extremely thin animal with no discernible body fat and nine indicated an extremely obese dog [16].

### 2.4. Fecal Sampling and Analysis

All fecal samples were collected by the same researcher (AO) either by digital removal from the rectum or by collecting feces that were observed being voided by the dog. Individual fecal samples were stored in either a 50-mL plastic specimen container or a 20-mL milk sample plastic container that was immediately placed in an insulated chiller bag with ice packs. At the end of the day, samples were placed in a refrigerator at 4 °C. Within ten days of collection, the chilled samples were transported to the laboratory for analysis using a same-day courier. When samples arrived, they were stored for a further one to 30 days in a refrigerator at 4 °C before a fecal parasitological examination was undertaken.

All fecal examinations were performed at Massey University Veterinary Parasitology Laboratory. The procedure used for the fecal examination was based on the flotation method described by Zajac et al. [17]. One gram of feces was weighed and placed into a sieve inside a 100-mL bowl. Approximately 20 mL of 33% ZnSO_4_ was poured onto the sample. The feces were disrupted, and coarse material retained by the sieve was discarded. The feces/ZnSO_4_ mixture was poured back into the test tube. Additional ZnSO_4_ was added, drop by drop, until the test tube was full and a positive (bulging over the top) meniscus had formed. A coverslip was placed onto the fluid meniscus and remained on when the tube was centrifuged at 200× *g* for five minutes. The coverslip was then transferred onto a microscope slide and examined with a compound microscope at 150× magnification. Ova and (oo)cysts were identified by morphology. When present, the ova of *Toxocara canis*, hookworms (either *Uncinaria stenocephala* or *Ancylostoma caninum*—the examination did not distinguish which species were present), and *Trichuris vulpis* were identified. *Sarcocystis* spp., sporocysts, the oocysts of *Cystoisospora*, spp., *Neospora caninum* or *Hammondia heydorni* (morphologically the oocysts are indistinguishable) and *Giardia* spp. cysts were identified when present. The fecal floatation method used was not expected to be able to reliably detect tapeworm eggs. Neither was the method deemed suitable for detecting *Cryptosporidium* spp. oocysts.

### 2.5. Classification of Dog Breed

Dogs in the study were categorized on their instinct for working stock rather than registered breeds. There were three categories of dogs: (i) huntaway; (ii) heading; and (iii) other. The huntaway is unique to New Zealand, and through selective breeding the dogs’ instinct to “hunt” or chase sheep away by barking has been enhanced. Huntaways are large dogs usually black and tan colored, and with coats that are smooth or long and shaggy. Heading dogs work silently and circle widely around the stock or cut one or more animals off from the others. Heading dogs will also be used preferentially to work with ewes during lambing. The majority of heading dogs are referred to as New Zealand heading dogs, which is not a registered breed. New Zealand heading dogs have been bred from the British Border collie, and like the huntaway, there is considerable variation in color, size, and coat. The third category “other” consisted of handy dogs, kelpie, and bearded collie. In New Zealand, handy dogs refer to a dog capable of doing both heading and huntaway work. The dogs may be a cross of the huntaway and heading dogs or other working dog breeds. Again, handy dogs have highly variable phenotypes.

### 2.6. Statistical Analysis

The number and percentage of dogs were determined and stratified by age (<2/2 to 3/4 to 7/≥8), sex (male/female), neuter status, breed, and BCS (2/3/4/≥5). Housing and preventive health regime did vary for dogs owned by the same person, and as such, the number and percentage of owners and dogs in each category were calculated. Separate two-way tables were constructed to explore the relationship between BCS (categorized as ≤3, 4, and ≥5) and age, sex, breed, and neuter status. Statistical significance was assessed using Chi-squared or Fisher’s exact if the actual or expected cell count was less than five.

The percentage of dogs infected with nematode or protozoan parasites was determined, and 95% confidence intervals were determined for “any parasite” and separately for *Toxocara*, hookworm, *Trichuris*, *Sarcocystis*, *Isopora*, and *Neospora/Hammondia*. Separate two-way tables were constructed to explore the relationship between infection with any parasite and sex, BCS, and frequency of use of anthelmintic drugs. Statistical significance was assessed using Chi-squared or Fisher’s exact as required. For *Toxocara*, hookworm, *Trichuris*, *Sarcocystis*, *Cystoisospora*, *Neospora/Hammondia*, and *Giardia* separate analyses were conducted to determine if there was an association between age and percentage of dogs with parasite lifecycle stages present in their feces and statistical significance was assessed using the Fisher’s exact test.

R Version 3.4.1 was used for all analysis, and the level of significance was *p* < 0.05.

## 3. Results

A total of 56 dog owners and 202 sheepdogs were enrolled in the study: 12 owners and 19 dogs were recruited at the 2010 North Island Dog Trial Championship, and the remainder were selected by the researcher from in the central North Island of New Zealand (Figure 1). The majority of dogs enrolled in the study were hunataway (51%) or heading dogs (49%), sexually entire (91%), and had a BCS of ≥4 (54%; Table 1). There was no significant association between BCS and age (*p* = 0.57), sex (*p* = 0.15), neuter status (*p* = 0.20), and breed of dog (*p* = 0.11).

The majority of owners kept their dogs in unmovable kennels that were raised off the ground (Table 2). All owners had the same preventative health regime for all the dogs under their care except one owner who had two of his dogs vaccinated annually and two that were not vaccinated. Of the dogs that we had preventative health data for, 53% of dogs were vaccinated at least every two years (104 of the 198 dogs; Table 3), and all dogs were reportedly given anthelmintic drugs, although 18% (36 of 196 dogs; Table 3) were treated less frequently than every three months. Seventy of the 196 dogs were not given any flea treatment, although six owners with a combined total of 21 dogs said they controlled fleas through water (e.g., swimming the dogs). Forty-four owners answered that they never or occasionally visited a veterinarian (Table 3). Among the group of owners who never or occasionally took their dog to a veterinarian, 14 owners indicated they vaccinated their dogs annually, and five owners reported vaccinating their dogs every second year but did not select “at vaccination” from the options for frequency of veterinary attention.

Owner’s reported feeding all their dogs the same diet, and the majority fed their dogs a combination of commercial dog food and home killed meat (41/53; 77%). Seven owners fed only commercial dog food, five fed only home killed meat, and one person said that they fed their dog food other than commercial dog food or home killed meat. Forty of the owners that indicated that they fed any home killed meat to their dogs reported they always froze it for more than seven days (33 of 46 owners; 72%), boiled it for more than hour (1 of 46 owners; 2%) or either froze or boiled the meat (4 of 46 owners; 9%). Another owner reported that they sometimes boiled the home kill before feeding the meat to their dogs. Three other owners who fed their dogs home killed meat said they never froze or boiled the meat before feeding their dogs. Only two of the 53 owners that provided data on the frequency of feeding, fed their dogs less than once a day which meant 187 of the 191 dogs were fed daily.

Fecal samples were collected from 171 dogs and 69, or 40% (95% CI 33% to 48%), were positive for parasites: 43 dogs had only one species of parasite, 21 dogs had two species of parasites present, and five dogs had three species of parasites present. Table 4 describes the proportion of dogs that were infected by each type of nematode or protozoan. There was no association between BCS and the presence of nematodes or protozoa (*p* = 0.64), sex of the dog (*p* = 0.07), type of flooring in the kennel (*p* = 0.25) or the frequency with which anthelmintic drugs were administered (*p* = 0.61). The percentage of dogs with parasite lifecycle stages present in their feces decreased with age for both *Toxocara* and *Giardia* (Figure 2). However, there were no significant associations between age and likelihood that hookworm (*p* = 0.99), *Trichuris* (*p* = 0.82), *Sarcocystis* (*p* = 0.35), *Cystoisospora* (*p* = 0.94) or *Neospora/Hammondia* (*p* = 0.07) would be present in the feces.

## 4. Discussion

This study of working farm dogs on the North Island of New Zealand found that 40% of dogs had parasite lifecycle stages present in their feces, with the majority of dogs having only one species of parasite present. The most commonly occurring parasite was *Sarcocystis* spp, which affected one in five dogs. Approximately, 20% of dogs had nematode eggs in their feces, with hookworm eggs being the most likely. The findings related to nematodes are similar to those reported in a previous study of working farm dogs in the North Island of New Zealand that reported the prevalence of nematodes was 21%, with hookworms also being the most common [18]. In comparison, a study of dogs in the South Island of New Zealand found an overall prevalence of 22%, but infections with *Toxocara canis* were the most common [19].

A much higher level of infection was reported in a 1981 study that involved post-mortem examination of the gastrointestinal tracts of 55 6–18 month old dogs, and showed an overall prevalence of nematodes and cestodes of 69% [10]. The latter study most likely utilized dogs surrendered to shelters, and therefore, were less likely to have received anthelmintic treatment, hence the higher prevalence. Nevertheless, other studies have shown that reliance on examination of feces for eggs will underestimate the true prevalence of nematode infections [20], however, not by a substantial margin. Other, alternative methods such as ELISA-based coproantigen detection assays [21], do not provide the breadth of coverage that the simple fecal floatation test does, and in any event, were not available when the study was conducted and would likely have been prohibitively expensive had they been.

The method used in the present study was not expected to reliably detect cestode eggs and indeed none were found, but nor were any segments detected in the submitted faucal samples. This may not be surprising given the likely regular use of cestocidal anthelmintics in these dogs. It is of interest that one of the authors (IS) recently dealt with a dog seen to be shedding tapeworm segments (later confirmed to be *Taenia pisiformis*), and fecal floatation examination of the dog’s feces yielded large numbers of taeniid eggs (unpublished observation).

Younger animals had a significantly higher prevalence of *Toxocara canis* and *Giardia* spp. The findings are not unexpected as *Toxocara canis* can be transmitted to puppies via the placenta and mammary glands, and patent infections are generally much less prevalent in older animals [7,22,23]. The lower prevalence of *Giardia spp.* in older dogs may be due to the development of protective immunity [24].

More than 80% of the dogs were reportedly given anthelmintic drugs at least every three months; however, giving drugs at this frequency did not eliminate infection. It is possible the way we asked the question contributed to this finding as farmers were only asked: “How often do you worm your dog(s)?” The questionnaire did not capture information about the products used and dose administered, which was an omission. In addition, dogs may not have been given sufficient product for their weight. Secondly, treating every three months may not eliminate the potential for egg shedding, and reinfection may occur as the minimum pre-patent period for *Toxocara canis*, hookworms, and *Trichuris vulpis* is 28 days, 14 days, and ten weeks, respectively [22,25]. The questionnaire did not collect any data on time since last treatment, and the presence of parasites could simply be because retreatment was now required. Alternatively, the owners may not be using an anthelmintic that targets nematodes, as farmers may instead use praziquantel on its own due to the importance of killing cestodes for sheep measles control. While there is ambiguity in the question, the results suggest farmers should not rely on anthelmintic administration as the sole method of gastrointestinal parasite control in their working sheepdogs and research is needed to determine if current treatment regimens are appropriate. As a further comment, the manufacturers of most products used to treat nematode and/or cestode infections in dogs do not claim their products are effective against protozoan parasite infections.

The current survey found that over 80% of owners feeding raw meat either froze the meat for seven days or boiled the meat for over an hour. Those that did not freeze or boil their meat are not adhering to the voluntary program to control Sheep Measles, and as such, risk exposing their dogs to *Taenia ovis* [26]. It was not clear if the failure to treat meat was due to non-compliance with the program or an active decision not to participate in the program. Either way, nearly 20% of farmers are not following this rather easy step to control disease on their farm that could impact their profits if sheep or lamb carcasses are downgraded or condemned [27]. Further research should be undertaken to explore uptake and compliance with the Ovis Management plan. It is noteworthy that the question specifically asked about meat rather than offal because there is still a legal requirement in New Zealand that offal from home killed animals be frozen for at least seven days or boiled for an hour as part of the country’s ongoing control of *Echinococcus granulosus* to cook offal for 30 min prior to feeding it [28]. Furthermore, even when owners froze meat at −10 °C for at least seven days, other parasites, like *Sarcocystis* spp., may survive [29].

The current study found that over 90% of dogs had a BCS score of less than five out of nine and nearly half of the dogs had a score of less than four. Further, there was no relationship between BCS and presence of parasites, age, breed or neutering status. A BCS of four and five are ideal and less than four are classified as “under ideal” [30]. We would caution against considering the BCS of less than four as less than ideal as research in BCS has focused on companion animals. It is entirely plausible that an ideal BCS for working dogs will differ to those of pets.

The current study reported more than 95% of owners fed their dogs once a day and nearly 80% of owners fed their dog’s commercial dog food and home killed meat. While the frequency of feeding was virtually identical to results presented in earlier work conducted by Singh and colleagues [13], the studies differ in the proportion that fed a mix of home killed meat and commercial dog food. Only half of the owners in the study by Singh and colleagues [13] fed a combination of home killed meat and commercial dry food during the year. The differences could reflect differences in the study populations, as the current study includes a range of different farmers, while the earlier survey by Singh and colleagues [13] surveyed people registered with the New Zealand Sheep Dog Trial Association and it is possible that the latter are more likely to feed commercial dry food. It is unlikely the difference is due to the fact that the current study focused on a single geographical area as the earlier study found no regional differences.

Approximately 80% of all owners reported that they never or only occasionally sought veterinary attention for their dogs, electing not to select the option “when vaccinated”. This is partially explained by the finding that nearly 30% of dogs were never vaccinated or vaccinated only as a pup. The reasons for the lack of veterinary involvement in the management of working farm dogs in this study is unclear. It is possible the dogs simply did not have any health problems and as such we would warn against concluding that working dogs would not receive veterinary attention if required. However, the study does highlight the problem with using veterinary clinic data as was done by Cave et al. [15] to gain a complete picture of the health of working farm dogs because the more serious traumatic injuries and diseases may be over-represented.

In conclusion, this study found high levels of infection with gastrointestinal parasites including nematodes in sheepdogs on farms in one region on the North Island of New Zealand, which is of concern for the health of the dogs and their owners, given the zoonotic potential of some parasites such as *Toxocara* and *Giardia*. In addition, that infection was prevalent despite most owners reporting that they gave anthelmintic drugs at least every three months (1) suggests owners should not rely on anthelmintic administration as the sole method of gastrointestinal parasite control in their working sheepdogs and (2) highlights a need for further research to determine if current protocols for anthelmintics are appropriate.

## Figures and Tables

**Figure 1 animals-09-00094-f001:**
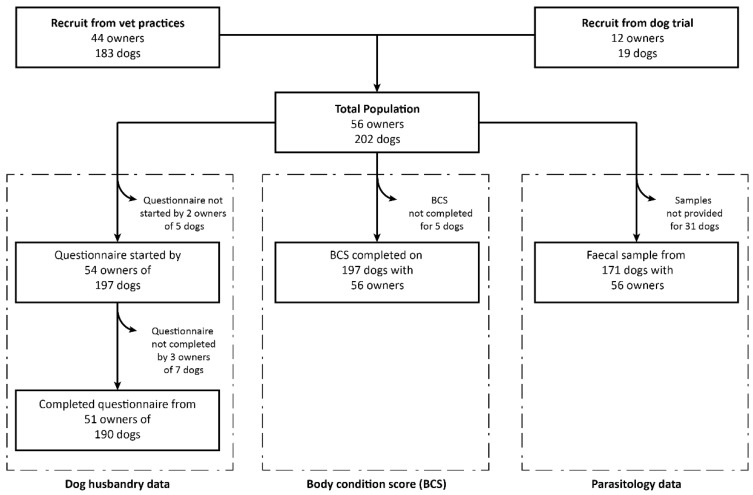
Flowchart describing the recruitment of participants in a cross-sectional survey of working farm dogs in the central North Island of New Zealand.

**Figure 2 animals-09-00094-f002:**
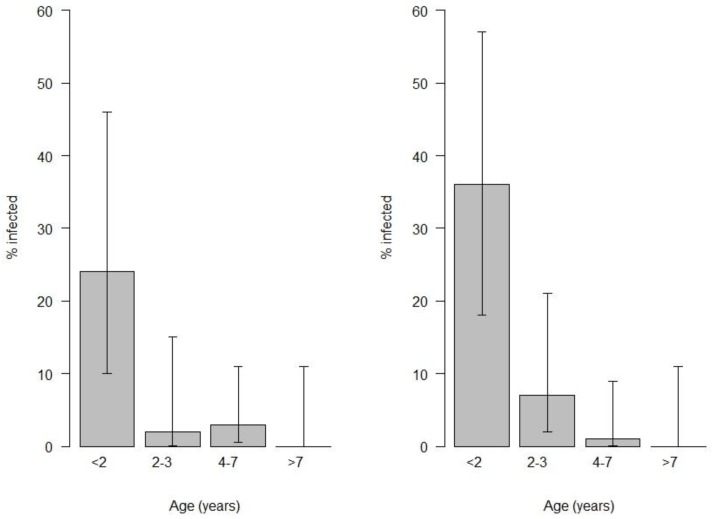
Percentage of dogs with fecal parasites decreased with age for *Toxocara canis* (**left**; *p* ≤ 0.0001) and *Giardia* (**right**; *p* ≤ 0.0001). Data from a cross-sectional survey of working dogs’ health conducted in the central North Island of New Zealand in 2010, which included 57 working farm dog owners and 202 working dogs. Note: Age was missing for four dogs owned by two different people and fecal samples were missing for 31 dogs owned by 19 different people. However, only one owner provided no fecal samples for any dog in their team.

**Table 1 animals-09-00094-t001:** Characteristics of working farm dogs (*n* = 197 dogs). Data from a cross-sectional survey of working dogs’ health conducted in the central North Island of New Zealand in 2010 which included 57 working farm dog owners and 202 working dogs.

Variable	Number	Percentage
**Age ^a^**		
<2	31	16
2 to 3	51	26
4 to 7	75	38
≥8	41	21
Breed ^a^		
Heading	89	45
Huntaway	101	51
Other	9	5
Sex ^a^		
Male	118	59
Female	83	41
Sexually entire		
No	18	9
Yes	173	91
Body condition score ^a^		
2	25	13
3	65	34
4	91	47
≥5	13	7

Note: Totals of percentages are not 100 for every characteristic because of rounding. ^a^ Data were missing for age (4 dogs), breed (2 dogs), sex (1 dog), neuter status (11 dogs), and body condition score (5 dogs).

**Table 2 animals-09-00094-t002:** Characteristics of housing for owners and working dogs from 53 owners and their 196 dogs. Data from a cross-sectional survey of 54 owners that completed a questionnaire about husbandry and management of their dogs.

Variable	Owner		Dogs	
	Number	%	Number	%
Frequency kennel is moved ^a,b^				
Not moveable	34	65	124	64
More than once a year	4	8	9	5
Less than once per year	4	8	14	7
Never	14	27	48	25
Floor design ^c,d^				
Raised slats	42	79	164	84
Raised solid	10	19	26	13
Dirt	4	8	6	3

Notes: Totals for percentages for the owners are not 100 for every characteristic because (1) some owners had different practices for their dogs and (2) because of rounding. Totals for percentages for the dogs are not 100 for every characteristic because of rounding. ^a^ Two owners, each with one dog, did not indicate how frequently the kennel was moved. ^b^ Four owners did not move all their dog’s kennels with the same frequency. ^c^ One owner with one dog did not given information about the type of flooring in the kennel or the frequency with which the kennel was cleaned. ^d^ Three owners did not have the same flooring in all their dogs’ kennels.

**Table 3 animals-09-00094-t003:** Characteristics of preventive health regimes for owners and working dogs from 54 owners and their 196 dogs. Data from a cross-sectional survey of 54 owners that completed a questionnaire about husbandry and management of their dogs.

Variable	Owner		Dogs	
	Number	%	Number	%
Frequency of vaccination ^ab^				
Only when pup	10	19	24	12
Annually	19	36	78	40
Every two years	6	11	26	13
Sporadically or unsure	11	21	26	13
No vaccination	10	19	42	21
Frequency anthelmintic drugs are administered ^a^				
Every month	2	4	5	3
Every 2 months	20	38	69	35
Every 3 months	22	42	86	44
Every 4 to 6 months	7	13	26	13
Annually	2	4	10	5
Type of flea treatment ^a^				
None	19	36	70	36
Commercial dog product	31	58	115	59
Commercial non-dog product	3	6	11	6
Frequency of veterinary visits ^c^				
At vaccination	7	13	32	17
Frequently	2	4	8	4
Occasionally	36	68	120	63
Never	8	15	31	16

Notes: Totals for percentages for the owners are not 100 for every characteristic because some owners had different practices for their dogs and because of rounding. Totals for percentages for the dogs are not 100 for every characteristic because of rounding. ^a^ One owner with one dog provided no information about the frequency of vaccination, the frequency with which anthelmintic drugs were given, and the type of flea control used. ^b^ One owner vaccinated two of their dogs annually and never vaccinated their other two dogs. ^c^ One owner with six dogs provided no information about the frequency of veterinary visits.

**Table 4 animals-09-00094-t004:** Number and percentage of dogs, with 95% confidence interval (95% CI), with fecal parasite by species of parasite. Data from a cross-sectional survey of working dog health conducted in the central North Island of New Zealand in 2010 in which fecal samples were collected from 171 dogs owned by 56 different people.

Species	Number	% (95% CI)
Neospora	4	2 (0–5)
Trichuris	8	5 (2–8)
Toxocara	9	5 (2–9)
Cystoisospora	10	6 (2–9)
Giardia	13	8 (4–12)
Hookworm	21	12 (7–17)
Sarcocysts	35	21 (15–27)
**Any parasite**	**69**	**40 (33–48)**

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
