# Peer review of "Parasitic Nematode and Protozoa Status of Working Sheepdogs on the North Island of New Zealand"

_animals, 2019, doi:10.3390/ani9030094_

Round 1

Reviewer 1 Report

General comments:

Overall, the study provides worthwhile findings to a limited audience (new Zealand sheep and cattle farmers). The scientific value and depth of the manuscript could be improved by including molecular analyses to positively identify parasite species, and/or include some of the data that was collected but not presented (eg blood samples). The inclusion of haematology and biochemistry data (eg. eosinophils and Hb) could provide greater clarity and add depth to the manuscript. This additional information could  help to provide a fuller interpretation of the animal's overall health and in particular, in relation to the parasite infection. 

The presence of faecal parasites alone is not indicative of parasite burden (quantitatively). Please remove this term (parasite burden) and replace with more accurate/appropriate wording throughout the manuscript. Please be consistent and use correct species/genus nomenclature throughout.

Specific comments:

Abstract:

L23 - insert 'may' before have

L30 (here and elsewhere) 'risk of infection' replace with 'prevalence of faecal parasites' or similar. The risk of infection was not determined and it cannot be said that it decreased with age.

L32 - insert 'reportedly' at the end of this line (after 'drugs were')

L 35 - as above - include 'reportedly administered'

Abstract should also include a sentence about the limitations of the study.

Introduction:

Could be improved - include additional information about types of parasites known to be present in NZ, and what types of laboratory techniques can be used to identify these, including limitations to techniques - especially the techniques that you chose for the study.

Materials:

Figure 1 needs better resolution

L78 - end of sentence - add 'participation' ?

L101 - example should be examination - there are other places where words have been omitted - please do another proof read.

L105 - briefly explain the scale - from 1 (emaciated) to 9 (obese) or similar description.

L121-122 This is confusing. It sounds as though the tube was centrifuged with the coverslip on?

Results:

The frequency of cleaning - more or less than once a year, is a very blunt measure, and not insightful or useful. Should probably not be included in the analysis or results.

L213 & 217 - reword 'risk of infection' - as explained earlier. Here and elsewhere.

Conflicting statements presented at L 214 and 219.

Figure 2 - presents data pooled for all parasites - this is not valid. Break the bar graphs up into sections for each parasite - or remove figure.

Discussion

L240 - 'had parasite stages' - reword

L257 - 'were supposed to be' - reword 'reportedly' ?

L283 - did you find any evidence of Echinococcus granulosus in your faecal samples? Comment on this?

L297 - this paragraph needs to ne shorter. more concise.

L323-5 - Remove last sentence - it has no relevance to the research.

Author Response

We wish to thank the reviewer for their time we appreciate the useful comments and suggestions and believe that the paper is of much higher quality as a result. Attached is a word document that addresses each of the issues raised.  

Reviewer 2 Report

animals-450112

Parasite status of working sheepdogs on the North Island of New Zealand

Minor revision

General comments:

The manuscript describes the examination of working sheepdogs for intestinal nematodes and protozoa. The topic is of great importance and merits publication. Some of these parasites are zoonotic and risk assessment strategies only can base on investigations like this.

One drawback is, that although the cestodes are discussed it is not stated whether they have not been looked for or they were not found (similar to any trematodes). If such data exist, it would be very beneficial to include them in the manuscript and name it “Intestinal parasite status of ….”. If not, I would suggest to change the title to “Status of parasitic nematodes and protozoa …..” or similar. Parasite status also would include blood parasites etc., which was excluded in the manuscript.

Detailed comments:

Title: Either change to “Intestinal parasite status of ….” And include cestodes, trematodes etc., or “Status of parasitic nematodes and protozoa …..”

L126: It should be “Cystisospora” instead of “Isospora”

L128: If possible include cestodes, trematodes etc. to widen the investigation to really intestinal parasite status.

L157: it is “Neospora” instead of “Neosporia”

L212and L218: some formatting error

L299,300,303 314: I would prefer using “… and coworkers”, “… and colleagues” instead of “..et al.”

Author Response

We wish to the thank the reviewer for their time and useful suggestions. Please find attached a point by point response to the issues raised.

Round 2

Reviewer 1 Report

I am satisfied that the authors have improved the manuscript to be acceptable for publication, however, some typographical errors remain and should be corrected prior to publication: 

L31 - word missing between 'not' and 'BCS'?

L51 - "on monthly" - delete 'on'?

L71 - delete "(eg. frequency kennels are cleaned) as this has now been removed from the analyses.

L90 - remove 'of' after flowchart

L308-9 - "to cook offal for 30 min...[30]" this statement is sitting on its own - remove or correct.